# What is the prevalence of COVID-19 detection by PCR among deceased individuals in Lusaka, Zambia? A postmortem surveillance study

Christopher J Gill [iD],[1] Lawrence Mwananyanda,[1] William B MacLeod [iD],[1] Geoffrey Kwenda,[2] Rachel C Pieciak,[1] Lauren Etter,[1] Daniel Bridges,[3] Chilufya Chikoti,[4] Sarah Chirwa,[4] Charles Chimoga,[4] Leah Forman,[5] Ben Katowa,[6] Rotem Lapidot,[7] James Lungu,[4] Japhet Matoba,[6] Gift Mwinga,[4] Benjamin Mubemba [iD],[8] Zachariah Mupila,[9] Walter Muleya,[10] Mulenga Mwenda,[11] Benard Ngoma,[4] Ruth Nakazwe,[12] Diana Nzara,[4] Natalie Pawlak,[13] Lillian Pemba,[4] Ngonda Saasa,[14] Edgar Simulundu [iD],[6] Baron Yankonde,[4] Donald M Thea[1]

For numbered affiliations see end of article.

**Correspondence to**
Christopher J Gill; cgill@bu.edu

## ABSTRACT

**Objectives** To determine the prevalence of COVID-19 postmortem setting in Lusaka, Zambia.

**Design** A systematic, postmortem prevalence study.

**Setting** A busy, inner-city morgue in Lusaka.

**Participants** We sampled a random subset of all decedents who transited the University Teaching Hospital morgue. We sampled the posterior nasopharynx of decedents using quantitative PCR. Prevalence was weighted to account for age-specific enrolment strategies.

**Interventions** Not applicable—this was an observational study.

**Primary outcomes** Prevalence of COVID-19 detections by PCR. Results were stratified by setting (facility vs community deaths), age, demographics and geography and time.

**Secondary outcomes** Shifts in viral variants; causal inferences based on cycle threshold values and other features; antemortem testing rates.

**Results** From 1118 decedents enrolled between January and June 2021, COVID-19 was detected among 32.0% (358/1116). Roughly four COVID-19+ community deaths occurred for every facility death. Antemortem testing occurred for 52.6% (302/574) of facility deaths but only 1.8% (10/544) of community deaths and overall, only ~10% of COVID-19+ deaths were identified in life. During peak transmission periods, COVID-19 was detected in ~90% of all deaths. We observed three waves of transmission that peaked in July 2020, January 2021 and ~June 2021: the AE.1 lineage and the Beta and Delta variants, respectively. PCR signals were strongest among those whose deaths were deemed 'probably due to COVID-19', and weakest among children, with an age-dependent increase in PCR signal intensity.

**Conclusions** COVID-19 was common among deceased individuals in Lusaka. Antemortem testing was rarely done, and almost never for community deaths. Suspicion that COVID-19 was the cause of deaths was highest for those with a respiratory syndrome and lowest for individuals <19 years.

## STRENGTHS AND LIMITATIONS OF THIS STUDY

⇒ Decedents were enrolled based on an a priori sampling scheme to minimise selection biases.

⇒ Contemporaneous death by age strata data for the whole deceased population were obtained to infer whether our sample was representative of the larger whole.

⇒ Where possible, samples underwent genetic sequencing, to overlay shifts in the prevalence of viral variants (Beta and Delta) with the prevalence of COVID-19 in the morgue.

⇒ Establishing causality based on sampling of the nasopharynx has limitations.

⇒ Our data all come from one city in one country, so we cannot directly generalise our results to other locations in Africa.

## INTRODUCTION

Despite the evident harm from the COVID-19 pandemic globally, accurate data about the pandemic's impact in Africa remain sparse. A recent multisite data identified the severe impact of COVID-19 in terms of deaths in African intensive care units. However, the ICU population is not broadly generalisable, and such observations do not yield a direct estimate of the overall prevalence of COVID-19 or its impact at a population level.[1] In early 2021, our team reported results from systematic COVID-19 postmortem surveillance conducted from June to October 2020 at the central morgue in Lusaka, Zambia.[2] Using PCR, we detected COVID-19 in nearly 20% of decedents, most of whom had died in the community. In that group, none had been tested for COVID-19 prior to death.

Yet, even among facility deaths, antemortem testing was uncommon. Hence, COVID-19+ deaths appeared rare in Lusaka because testing was rarely done in life. We theorised that the so-called 'Africa Paradox'[3–8] was a myth born from insufficient surveillance data. Other groups have since reached similar conclusions.[9–12]

Our analysis yielded other provocative findings. In contrast with the USA and other high-income country (HIC) settings where deaths are concentrated in the elderly, COVID-19+ deaths in Lusaka occurred more evenly across the age spectrum. Furthermore, 10% of COVID-19+ deaths occurred in children, a surprising finding given how rare paediatric COVID-19 deaths are in HICs. It remains an open question whether these paediatric deaths were caused by COVID-19 or whether the virus was a coincidental finding.

These conclusions were conservative given that our analysis rested on only 70 COVID-19+ decedents. Subsequently, our team conducted a second round of postmortem COVID-19 surveillance, spanning the period January through June 2021. In this longer and larger study, we again quantified the burden of fatal COVID-19 by setting (community vs facility deaths), calendar date and geography; and observed the impact and patterns of clinical presentation among children versus adults. In addition, we aligned our prevalence data with genetic sequencing to characterise shifts in viral lineages/variants over time.

## METHODS
### Overview
Our COVID-19 surveillance work builds on a larger body of postmortem surveillance work initiated to define the fatal burden of Respiratory Syncytial Virus and *Bordetella pertussis*. With the onset of the pandemic, the presence of this established infrastructure and team allowed us to quickly amend our protocols to shift from an exclusive focus on RSV and pertussis in infants to measure COVID-19 prevalence across all ages. The RSV results have been published,[13–15] and the pertussis results are currently under review.

We direct the readers to our prior publication for complete details of the surveillance methods.[2] Throughout this paper, we refer to several rounds of surveillance. Round 1 ran from June to October 2020; round 2, comprising the new data in this paper, ran from January to June 2021. Round 3 is ongoing, was initiated early in 2022, and is planned to run through January 2023. These surveillance windows were the solely due to funding and not based on assumptions about COVID-19 prevalence over time.

The research was conducted with the approval of the ethical review boards from Boston University (reference number H-36469) and the University of Zambia (reference number 015-11-18); written informed consent was obtained from the next of kin or family representative in all cases.

### Sampling strategy
When capturing a high proportion of all deaths, postmortem surveillance is robust against ascertainment biases and provides a direct and unambiguous measure of a disease's prevalence, though inferring causality due to the disease may still be challenging. Given concerns about viral RNA degradation leading to false negative results, enrolment was restricted to deaths within the preceding 48 hours.

We enrolled deceased individuals at the University Teaching Hospital (UTH) morgue. UTH is the main teaching hospital for the University of Zambia Medical School. Its morgue is the largest in the city and where ~80% of decedents are passed on to their families, irrespective of socioeconomic status of the decedent. This, plus the legal requirement to obtain a burial certificate before interring a body, most of which are issued at the UTH morgue, makes it highly representative of all deaths in Lusaka.

Due to the high volume in the morgue and finite capacity of our team, we capped enrolments at ~5–6 deaths per day. Also due to high volume, we enrolled community deaths at a 1:3 ratio, while enrolling the less common facility deaths at a 1:1 ratio. In response to the surprising high proportion of paediatric deaths in surveillance round 1, we elected to also oversample this group enrolling infants (<1 year) at a 1:1 ratio. This required that we adjust for these ratios in our total population prevalence estimates (see statistical analysis). Our team had no access to the clinical data about each decedent and could not know the PCR results prior to consent and enrolment. Therefore, we were confident that our enrolment strategy was robust against selection biases with respect to COVID-19 status.

Concurrently, we extracted total deaths by age stratum from the official government burial registry. This allowed us to estimate the contemporaneous proportion of all deaths represented by our sample, and, by comparing the age by death distribution of the two groups, assess whether the enrolled sample was representative of the total.

### Data collection and case definitions
We defined a 'facility death' as one that occurred under care at UTH or a referring facility. We defined a community death as one that occurred outside of medical care. We defined paediatric deaths as those occurring between 0 and ≤19 years, recognising that some may choose to define the end of adolescence differently, and that such definitions may not align with official age of emancipation laws in different parts of the world.

Following enrollment, individual and household demographic data were collected on all decedents, along with any antemortem COVID-19 testing results that were documented in the medical chart (for facility deaths) or reported by the next of kin/representative of the decedent during the verbal autopsy (for community deaths).

The most difficult challenge was inferring the causal role of COVID-19 in each death: the SARS-CoV-2 virus may have a direct or indirect role in deaths or could be a coincidental finding. The US CDC has issued guidance stating that in deaths where COVID-19 is detected, the virus should be assumed to be the direct or underlying cause of death, absent exonerating circumstances.[16] That notwithstanding, to better infer the causal role of COVID-19 in deaths, we used the clinical data describing symptoms during the fatal event to categorise the syndromic presentation of each death. For facility deaths, these data were obtained from a medical chart extraction focusing on symptoms at initial presentation to UTH; for community deaths, we conducted a verbal autopsy using the abbreviated tool validated by the Institute for Health Metrics Evaluation (IHME).[17] From these syndromic data, using the same approach as in our initial publication,[2] we classified COVID-19 PCR+ adult decedents (>19 years) into the following categories:

1. Probable COVID-19 deaths, where the individual had presented with any combination of fever and/or respiratory symptoms.
2. Possible COVID-19 deaths, where the individual presented without classic respiratory symptoms, but rather with known sequalae of COVID-19 suggestive of a heart attack, stroke or other acute vascular event related to COVID-19's propensity to increase the risk of clotting disorders.[18]
3. Probably not COVID-19 deaths, where the individual had an exonerating cause of death suggesting that the COVID-19 finding on PCR was co-incidental.
4. Unknown, where the data were insufficient to adjudicate.

The reasons we only applied this strategy to adults is that this taxonomy quickly proved ill-suited for younger individuals. Unlike adults, where fever and/or respiratory symptoms predominated, the paediatric presentations were more variable and respiratory symptoms less common. Since we were (and remain) uncertain about how COVID-19 should present clinically in children, we worried that creating a priori case definitions based on how the virus behaves in adults could be misleading. Similarly, applying a new definition developed iteratively from the constellation of symptoms observed in children and then applying that definition forwards without validation against an external gold standard (such as histopathology coupled with PCR), constitutes a logical circularity.

Facing this uncertainty, we opted simply to describe the paediatric case presentations without attempting to infer causality for decedents aged≤19 years.

## Biological sample collection

From each deceased individual, we obtained posterior nasopharyngeal (NP) samples. We used flocked-tipped NP swabs (Copan Diagnostics, Murietta, CA, USA),[19 20] inserting the swab into both nares and then rotating 180 degrees in both directions and collected into 3 mL vials of universal transport media. Samples were stored cold or on ice until processing at our onsite molecular laboratory.

## Laboratory procedures
### PCR testing for COVID-19

After vortexing to elute samples, total nucleic acid for PCR was extracted using the EasyMAG system (Biomerieux, Marcy L'Etoile, France).[21] We ran PCR for 45 cycles using the US CDC's COVID-19 testing kit, which targets the N1 and N2 nucleocapsid proteins.[22] We also ran PCR on each sample against the constitutive human enzyme RNAseP. The presence of RNAseP indicates that the NP swab made effective contact with the respiratory mucosa and that there were no PCR inhibitors.

As per the instructions in the CDC's protocol, we defined a COVID-19 test result as positive if the PCR cycle threshold (Ct) was <40 for both the N1 and N2 genes, provided that each assay demonstrated a logarithmic fluorescence amplification curve, had RNAseP detectable at Ct<40 and had positive and negative plate controls performing as anticipated.

However, Ct is a continuous variable, and the selection of any Ct cut point is arbitrary and hence controversial.[23] While a lower Ct, reflecting a higher viral load, is often associated with increased clinical severity, most COVID-19 surveillance studies assess individuals when they present acutely. In the acute setting, viral loads, and hence PCR signals, are likely at peak intensity. By contrast, the population in this study were all tested after they had died, and currently there are few data about what PCR signals should look like at the end of an arc of infection. At that point, the viral load and PCR signal intensity could have declined (or be undetectable), even if COVID-19 had set in motion a set of events culminating in death. For this reason, we accepted a Ct<40 as positive, even though lower Ct values have often been used for defining COVID-19 status at the time of initial presentation. But since this substitutes one arbitrary cut point (<40) for another arbitrary cut point (<30), we also report the number of decedents with COVID-19 detectable between Ct>40 and ≤45, though we have not included these in our summary analyses.

### Genetic sequencing of selected samples

To characterise the distributions of novel COVID-19 variants, we conducted genetic sequencing to detect viral variants on the subset of samples with a Ct<30. Prior experience has shown that nucleic acid concentrations are rarely sufficient for sequencing above this threshold. For this, we tested samples both from surveillance rounds 1 and 2, describing shifts in dominant COVID-19 variants in Lusaka over the cumulative period of surveillance.

Total RNA was re-extracted from vortexed NP specimens using the QIAamp Viral RNA Mini Kit (Qiagen, Hilden, Germany) as prescribed by the manufacturer. Complementary DNA was synthesised using LunaScript RT SuperMix Kit (New England Biolabs, Ipswich, Massachusetts, USA) according to the manufacturer's protocol

and multiplex PCR was conducted using custom primers, which were designed using Geneious software version 10.0.9, as used previously to sequence the first COVID-19 case in Zambia using the Sanger method.[24 25] The multiplex PCR generated overlapping amplicons which were then sequenced on the MinION (Oxford Nanopore Technology, UK) platform. The samples were multiplexed using the Oxford Nanopore native barcoding expansion kits 1–12 and 13–24 in combination with the ligation sequencing kit 109 (Oxford Nanopore). The data generated through the MinION were processed using the standard ARTIC bioinformatic pipeline (https://artic.network/ncov-2019/ncov2019-bioinformatics-sop.html). After removal of sequencing primers, the consensus sequences were deposited in the GISAID database (https://www.gisaid.org/). PANGO lineages were then determined for each genome using Pangolin V.3.1.16 (https://pangolin.cog-uk.io/).

## Statistical analysis

Given the uncertainties of future detection rates for COVID-19, our sampling was not based on a priori power assumptions. Rather, we sought to enrol as many decedents as our team could reasonably accommodate. Our statistical analysis was a straightforward comparison of proportions that did not require statistical modelling or imputation. COVID-19 status, based on PCR, was stratified by setting (community vs facility deaths), by age groups, by calendar date and by geography (city ward).

Where indicated, we have adjusted prevalence estimates by calculating weighted outputs to account for the different sampling ratios used during enrolment. We calculated weights so that the sum of the weighted population would sum to the sampled population: unweighted sample (1118) = weighted sample (1118). This allowed us to calculate the relative frequency of our unweighted sample in the four weighted groups: (1) facility deaths <1 year, (2) all other facility deaths, (3) community deaths <1 year of age and (4) all other community deaths. We sampled the facility deaths <1 year, all other facility deaths and community deaths in infants <1 year at a 1:1 ratio and the community deaths ≥1 year at a 1:3 ratio. We then calculated the weighted sample by multiplying the inverse of the sampling ratio and calculated the relative frequency of the weighted sample. Setting the weighted sample size at the actual enrolled sample size of 1118, yielded sample weights of 0.53 for the portion of the sample that we recruited as 1:1 and the sample weight of 1.60 for the portion of the sample that we recruited as 1:3. Hence, the ratio of these sample weights, adjusting for rounding, is 1.60/0.53=~3:1. The reason for this weighting should be self-evident, since without such weighting the prevalence estimates will necessarily over or under estimate the true prevalence across different groups of decedents.

For analyses of COVID-19 deaths over time and place and for the genetic sequence data, we combined results from rounds 1 and 2 of surveillance. Effectively, this provides a time series analysis over a 1-year period minus the 3-month gap from October 2020 to January 2021 defined by the end and start of adjacent funding cycles. Where appropriate, we reference our results from round 1 to compare and contextualise results from round 2. For all statistical analyses and most data manipulations, we used SAS software (Cary, North Carolina, USA).

For the geospatial analyses, we used ArcGIS software (ESRI, Redlands, California, USA). We pulled population size data from the Zambia Data Hub: https://zambia-open-data-nsdi-mlnr.hub.arcgis.com/, which is managed by the Government of Zambia through the Ministry of Lands and Natural Resources, and the Zambia Statistics Agency. We downloaded these ArcGIS layers to the level of Lusaka's city wards. We then created a two-dimensional heat map that indexed the total number of COVID-19+ deaths against total enrolled deaths across each of Lusaka's main city wards.

## Patient and public involvement

The participants in this study were all deceased. There was no involvement of patients or the public in the design of this study. The results of this research have already been made publicly available via a posting on MedRXiv.

## RESULTS
### Study overview and summary of PCR results

This second round of surveillance spanned January–June 2021. The newly enrolled cohort included 1118 deceased individuals ranging in age from <1 to 102 years. Contemporaneously, 6270 deaths were reported in the Lusaka burial registries. Thus, our sample represented 17.8% of all deaths in this period. The age distribution of the enrolled versus the total deceased population were similar, arguing to representativeness (online supplemental figure S1).

PCR was successfully conducted on 100% of the NP swabs, and 1116 had an RNAseP<40, indicating adequacy of sample collection. COVID-19 was detected among 29.3% (327/1116) of decedents without weighting, and 32.0% (effective sample size=358/1116) after adjusting for weighting (table 1).

Thus, the overall prevalence of COVID-19 in round 2 was roughly double what we observed in round 1 (16%, 58/364). The median Ct values for the N1 and N2 targets were ~33 in both cases (online supplemental figure S2). We note that an additional 58 samples (45.8 weighted) had COVID-19 detectable at a Ct≥40–45 but were *not* included in subsequent analyses.

The weighted proportion of females among the COVID-19+ and COVID-19– groups was nearly identical (40.6%, 453/1116). By contrast, the COVID-19+ decedents were significantly older than those without COVID-19 (median 48 years (IQR 30–70 years) vs 39 years (IQR 0–58 years)), Wilcoxon Ranked Sum test p-value<0.001 (table 2)).

**Table 1** Postmortem COVID-19 PCR results, stratified by setting

| Setting of death | Negative, Ct≥40 | Positive, Ct<40 | Totals |
|---|---|---|---|
| Unweighted | | | |
| Community % (n/N) | 66 (358/543) | 34 (185/543) | 49 (543/1116) |
| Facility % (n/N) | 75 (431/573) | 25 (142/573) | 51 (573/1116) |
| Both settings combined | 71 (789/1116) | 29 (327/1116) | 100 (1116/1116) |
| Weighted | | | |
| Community % (n/N) | 65 528.2 | 35 282.7 | 72.7 811 |
| Facility % (n/N) | 75 229.5 | 25 75.6 | 27.3 305 |
| Both settings combined | 68% (757.7/1116) | 32% (358.3/1116) | 100% (1116/1116) |

Ct, cycle threshold.

## Proportion of deaths and COVID-19+ deaths by facility versus community setting

Within the 1116/1118 (99.8%) of total deaths with a valid test result, there were 573 facility deaths and 543 community deaths. While this implies that the ratio of facility to community deaths was similar, this does not account for the different enrolment ratios applied to each group. After weighting, 73% of all deaths occurred in the community versus 27% at a facility, for a ratio of ~three community deaths for each COVID-19+ facility death.

Similarly, COVID-19 was detected among 24.8% (142/573) of facility deaths and 34.0% (185/543) of community deaths. Considering the contribution of deaths by setting, among the 327 COVID-19+ decedents,

**Table 2** Basic demographics of the postmortem cohort, stratified by COVID-19 status

| | Negative, Ct≥40 | Positive, Ct<40 | All deaths |
|---|---|---|---|
| Demographic characteristic | (n=789) | (n=327) | (n=1116) |
| Unweighted percent females (n/N) | 42.3 (334/789) | 41.0 (134/327) | 41.9 (468/1116) |
| Weighted percent female (n/N) | 41.0 309/758 | 39.9 144/358 | 40.6 453/1116 |
| Unweighted median age at death, years (IQR) | 32.5 (0–54) | 44.0 (27–68) | 36.0 (1–58) |
| Weighted median age at death (IQR) | 39.0 (10–58) | 48 (30–70) | 41 (23–64) |

Ct, cycle threshold.

56.6% (185/327) were community deaths and 43.4% (142/327) were facility deaths. After weighting to adjust for enrolment ratios, community deaths accounted for 79% and facility deaths for 21% of total COVID-19+ deaths. Stated another way, for each COVID-19+ decedent detected at a facility, ~four COVID-19+ decedents were identified in the community.

## COVID-19 deaths by age group

During round 1, the proportion of COVID-19+ individuals was similar across all age groups and 10% of all COVID-19+ deaths were in children ≤19 years.

Round 2 yielded a similar pattern with COVID-19+ deaths occurring across all age groups and with relatively similar proportions out of total deaths in each group (see figure 1). After weighting, adults aged >19 years comprised 86.2% (300/358) of COVID-19+ deaths and children 0 to ≤19 years for 14.9% (53/358) (table 3).

Overall, 78.1% (208/358, weighted) of COVID-19+ decedents were aged <60 years. Contrasting the death by age distributions for the COVID-19+ decedents relative to the age distributions for the total population from the burial registries, the COVID-19+ deaths were relatively under-represented among children and relatively over represented in the elderly. Outside of these age extremes, the age by death distributions for the two cohorts were similar (online supplemental figure S1).

## Antemortem testing for COVID-19

During round 1, antemortem testing for COVID-19 was rarely done, occurring in only 10% of facility deaths and none of the majority of deaths that were in the community.

In Round 2, antemortem testing rates improved but only among facility deaths. Within this group, 52.6% (302/574) were tested antemortem, of whom 25 were reportedly positive (see online supplemental table S1). During this period, nearly all COVID-19 testing at UTH relied on rapid antigen tests. However, among the community deaths, we were still only able to document antemortem testing for 1.8% (10/544) of COVID-19+ decedents.

## Temporal shifts in COVID-19 prevalence, dominant variants and geography

Given these changes between round 1 and round 2, we further explored the relationship between total deaths, COVID-19 prevalence, viral lineages and geographic distribution in time series analyses. These results, combining rounds 1 and 2 data, are presented in figure 2.

Several features are noteworthy. First, the data collected in round 1 document what in hindsight was likely the first, and smallest, of three waves of COVID-19 to occur in Lusaka during our surveillance period (panel A). Noting the gap in surveillance (between adjacent funding cycles) from October 2020 to January 2021, wave 1 occurred in June–October 2020; wave 2 in January–February 2021 and wave 3 in May–June 2021. Round 2 surveillance ended in

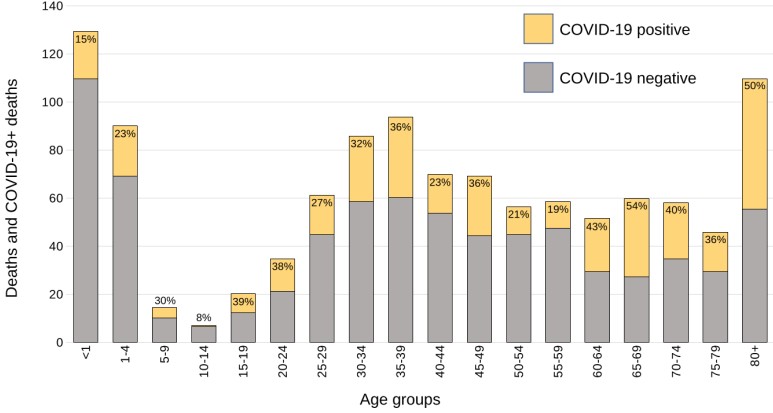

**Figure 1** Enrolled decedents and COVID-19+ deaths by 5-year age strata. Each stacked bar in the histogram shows the total number of enrolled decedents within each age stratum. The portion in yellow represents those who tested positive by PCR (Ct<40) for COVID-19 and those in grey those who tested negative (Ct≥40). The percentage of positives is indicated at the top of each column. The totals have been adjusted to account for differing enrolment ratios. Ct, cycle threshold.

June 2021, precluding observations about when wave 3 peaked and receded.

Second, during waves 2 and 3, the proportion of decedents who tested positive for COVID-19 peaked at 91.7% prevalence in January and 83.8% in June of 2021. Both represented a statistically significant increase over the average prevalence during rounds 1 and 2 (analysis of variance, p<0.001). The proportion of deceased who tested positive for COVID-19 by month are summarised in online supplemental table S2.

Third, each wave presented a distinct pattern in terms of prevalent viral lineages (panel B). Combining rounds 1 and 2 and selecting those with a Ct<30, we successfully sequenced 96 isolates. From these, lineage AE.1 was dominant during wave 1. During wave 2 in January–February, the B.1.351 variant (the Beta variant) was dominant. Beta

was supplanted by the emergence of the B1.617 variant (Delta) during wave 3, which accounted for 100% of all sequenced isolates during this period.

Concurrently, we mapped the distribution of COVID-19+ cases indexed against total deaths over time across Lusaka's major city wards (panel C). The areas shaded dark blue represent areas where COVID-19+ deaths were disproportionately high relative to total deaths, while those in light pink, light blue or white are areas where COVID-19+ deaths were under-represented relative to total deaths. Within this taxonomy, the highest burden of COVID-19 was in Lusaka's poorest and most densely populated areas that constitute the peripheral ring of peri-urban compounds to the city's north, west and south, and largely spared the more affluent neighbourhoods that cluster in the centre of the city.

### Clinical presentation of COVID-19 positive children and adolescents

Recalling that infants <1 year were intentionally oversampled, requiring weighting to draw overall prevalence estimates, most deaths in children and adolescents ≤19 years were in infants. With that caveat, online supplemental figure S3 summarises the syndromic presentations for the COVID-19 positive deceased children <19 years, stratifying these by age groups and community/facility settings. Since we are describing these on a per individual basis, there was no need to weight these results.

The syndromic categorisations are not mutually exclusive. Clinical presentations appeared to vary by age and setting. Among infants who died at a facility, respiratory symptoms and fever predominated, whereas fever, respiratory and gastrointestinal symptoms occurred in similar proportions among those who died in the community. Among children 1–4 years, fever and gastrointestinal symptoms were more common than respiratory symptoms. Very few COVID-19+ deaths were detected among children aged 5–14 years (n=5). Among children 16–19 years (n=5, all from the community), the only reported symptoms were fever and respiratory distress, consistent

**Table 3** Distribution of COVID-19 positive deaths by 20-year increments

| Age at death (years) | Ct≥40 (neg) | Ct<40 (pos) | Total | Ct≥40 (neg) | Ct<40 (pos) | Total |
|---|---|---|---|---|---|---|
| | Unweighted | | | Weighted | | |
| 0–19 | 308 82.8% | 64 17.2% | 372 | 205.6 79.4% | 53.3 20.6% | 261.0 |
| 20–39 | 159 66.5% | 80 33.5% | 239 | 184.8 67.1% | 90.5 32.9% | 275.3 |
| 40–59 | 171 73.4% | 62 26.6% | 233 | 190.1 74.8% | 63.9 25.2% | 253.9 |
| 60–79 | 108 58.1% | 78 41.9% | 186 | 120.3 55.9% | 94.8 44.1% | 215.1 |
| 80–99 | 42 52.5% | 38 47.5% | 80 | 55.4 52.0% | 51.1 48.0% | 106.5 |
| 100+ | 0 0.0% | 2 100.0% | 2 | 0 0.0% | 3.2 100.0% | 3.2 |
| Total | 788* | 324* | 1112* | 757.7* | 358.4* | 1112.1* |

Row results are N (top) and % (bottom) in each cell.
*Four had missing age data, of which three were COVID-19 positive and one was COVID-19 negative, for 789, 327 and 1116, respectively.
Ct, cycle threshold.

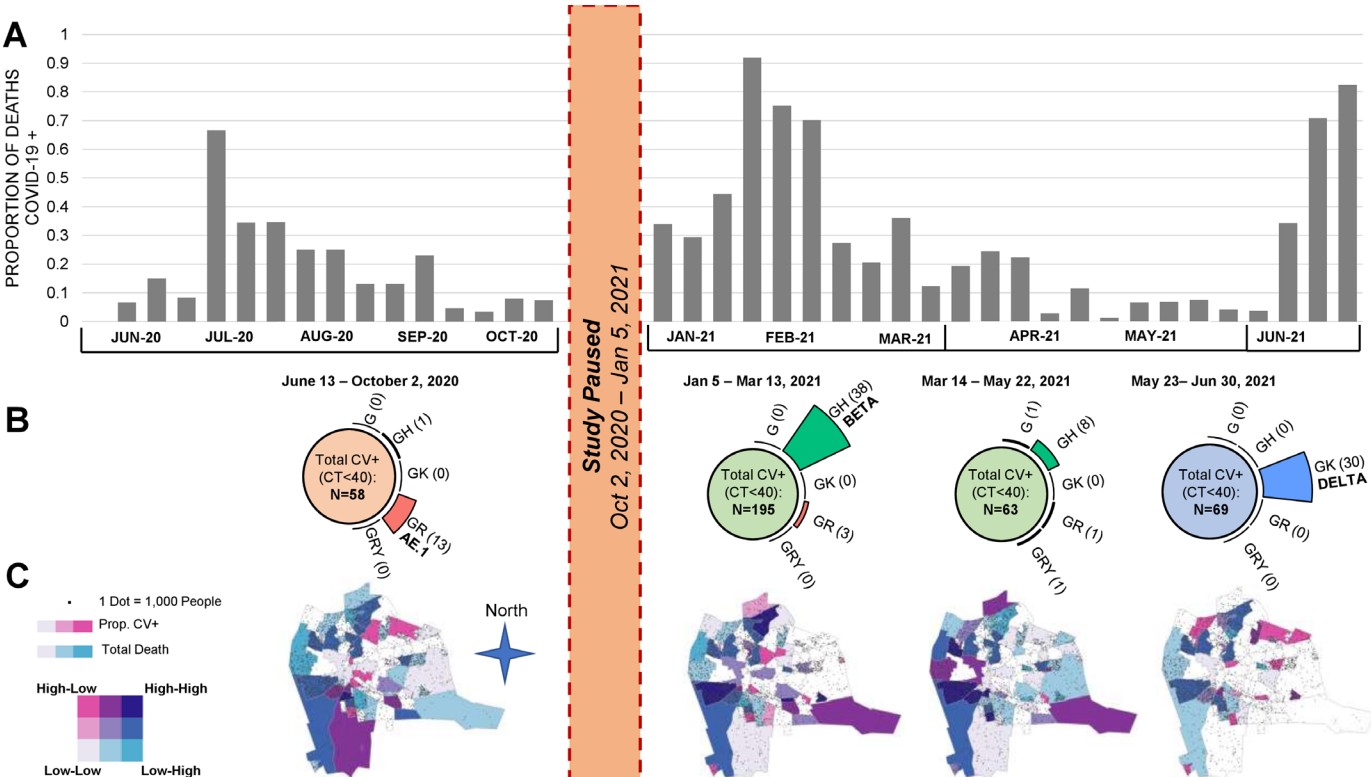

**Figure 2** Proportion of COVID-19+ deaths and viral lineages/variants, by time and geography. Panel (A) The histograms represent the total number of COVID-19 positive deaths identified during the two rounds of surveillance. The gap from October 2020 to January 2021 represents the end and start of funding cycles. It is apparent that there were at least three waves of transmission in 2020–2021. Since our enrolment strategy shifted between round 1 and round 2, we have adjusted the proportions for the round 2 data by weighting to account for the different enrolment ratios applied by age and setting. Panel (B) Combing the samples corresponding to each of the three waves, and the period in between waves 2 and 3, we clustered the viral lineages as shown on the top half of the figure. As can be seen, wave 1 was dominated by lineage AE.1, wave 2 by the 'beta' variant and wave 3 by the 'delta' variant. For these data, we present the results without adjustments for weighting. Panel (C) Over these same time intervals, we summarise the distribution of cases indexed against total population within the major city wards of Lusaka. The colours correspond to a heat map defined by the union of 'total deaths' in pink and 'COVID-19+' deaths in blue, and therefore provide an indexed metric of COVID-19 deaths against all deaths. Those with darker shading indicate that COVID-19 deaths were disproportionately high relative to all deaths; those that are lightly shaded or white are where COVID-19 deaths were disproportionately low, relative to total deaths. As can be seen, the highest concentration of cases clustered in the peripheral areas of the city, which correspond to the poorest and most densely populated parts of the city. Whereas the wealthier sections, clustering in the middle of Lusaka, were relatively spared. Round 2 results have also been adjusted for enrolment ratios as in panel A. Ct, cycle threshold.

with COVID-19's presentation in adults. In both community and facility settings, there were several children with no reported symptoms, though this may simply reflect a lack of documentation. Acknowledging the sparse data from older children, these results are broadly consistent with our observations in round 1 where we had noted the relative infrequency of respiratory symptoms and commonality of gastrointestinal symptoms, particularly among children ≤5 years. Online supplemental table S3 provides a detailed line listing summarising the clinical presentation of each COVID-19+ paediatric death by setting (facility deaths online supplemental table S3a, community deaths online supplemental table S3b).

### Inferring the causal role of COVID-19 in these deaths

During round 1, among those who had sufficient clinical data to allow an assessment of causality, nearly all were judged to have 'probably' or 'possibly' died from COVID-19.

In round 2, 89.3% (292/327) of deaths had sufficient data for clinical adjudication, leaving 10.7% (45/327) that could not be adjudicated. Excluding the paediatric cases and those with insufficient data for analysis, we were able to adjudicate 219 deceased adults. Of these, 155/219 (70.8%) were considered 'probably' or 'possibly' due to COVID-19, and 29.2% 'probably not' due to COVID-19. When adjusted for weighting, this combined proportion rose to 73.9% (181/244) (see table 4).

We recognised that the categories used for adjudication (probable, possible and probably not) rested on clinical information that may have been incomplete, a particular problem with the verbal autopsy data given that the respondent providing the information was necessarily

**Table 4** Causal role of COVID-19 among PCR(+) deceased adults*, >19 years

| Role of COVID-19 in death of patient | Median ct (IQR) | Cases (N) | Percent | Weighted cases (N) | Weighted percent | Cumulative total for cases that could be adjudicated | Cumulative percent (n/N) for cases that could be adjudicated |
|---|---|---|---|---|---|---|---|
| Probably due to COVID-19 | 31.6 (24.2–35.5) | 127 | 48.7 | 142.2 | 46.6 | 142 | 58.2 (142/244) |
| Possibly due to COVID-19 | 30.2 (23.2–35.7) | 28 | 10.7 | 38.3 | 12.6 | 180 | 73.8 (180/244) |
| Probably not due to COVID-19 | 34.9 (28.0–37.3) | 64 | 24.5 | 63.9 | 20.9 | 244 | 100 (244/244) |
| Insufficient data to adjudicate causality | 32.5 (23.9–35.5) | 42 | 16.1 | 60.7 | 19.9 | Not applicable | Not applicable |

*Clinical presentations of children with COVID-19 (n=63) were variable and often lacked typical symptoms of respiratory distress. Since we were uncertain how COVID-19 should behave in children, we did not attempt to adjudicate these cases in terms of presumed causality. The results above have been weighted to adjust for the different enrollment ratios.
Ct, cycle threshold.

relaying these data secondhand. A gold standard for asserting causality would have been PCR coupled with histology showing end organ damage consistent with SARS-2 infections. Indeed, we had previously published such pathology based on minimally invasive tissue sampling (MITS) among several of the cases here that had been diagnosed antemortem. As reported in that paper, these individuals showed evidence of diffuse organ injury to the lungs, kidneys and liver in patterns consistent with COVID-19 disease.[26] However, the identification of these decedents in life was unlikely done at random but rather based on clinical suspicion, creating the potential for ascertainment bias.

Given these limitations, we conducted additional causal inference analyses around the Ct data for the full cohort, which we felt would be more robust against selection bias. We reasoned that higher viral loads (and hence lower Ct values) should align with increased severity of disease. If so, we hypothesised that decedents deemed 'probable COVID-19', reflecting a high suspicion that COVID-19 was the cause of death, would on average have lower Ct values (reflecting higher viral loads) than those whose death was deemed 'probably not' COVID-19. The data in table 4 supported that hypothesis.

Among those whose deaths were adjudicated as 'probably' due to COVID-19, the median Ct values were 31.6 (IQR CI 24.2 to 35.5), whereas those adjudicated as 'probably not' COVID-19 had a median Ct value of 34.9 (IQR 28–37.3), or a ~10-fold increase in viral load. Thus, COVID-19+ individuals whose clinical data were most suggestive of a COVID-19 illness by presentation had the highest viral loads.

Following this reasoning further, we conducted an analysis exploring the distribution of Ct values by age groups. As noted previously, we were sceptical that COVID-19 was the cause of death in young children, largely because the clinical presentations in children, and especially children <5 years, differed from adults. Therefore, we reasoned that COVID-19 was more likely to be a coincidental finding in children. To test that contention, we stratified

the proportion of PCR results by age group sorting them into two groups: those with the strongest signals (ie, Ct<30) and those with weaker signals (ie, Ct 30≤Ct≤40). We then calculated the ratio of the proportions of weaker to strongest Ct values within each 20-year age stratum. We hypothesised that if COVID-19 was a coincidental finding, we would observe a higher ratio of weaker to strong results than if COVID-19 was the cause of death. The results shown in online supplemental table S4 supported that hypothesis. After adjusting for weighting, the ratio of weaker to stronger Ct results was 6.6:1 for those aged 0–19 years, compared with a ratio of 1.9:1 across all ages combined. With each increase in age stratum, the proportion of weak to strong PCR signals declined in a stepwise fashion, such that for those 80 years and older, the ratio was 1:1. We draw two conclusions from this analysis. First, the results supported our assumption that COVID-19 was likely a coincidental finding among children. Second, by extension, COVID-19 likely played a causal role in the deaths of older individuals.

## DISCUSSION

This second round of postmortem surveillance for COVID-19 reinforces and expands on our previous observations. First, even if we base our conclusions solely on those who were deemed 'probably due to COVID-19' based on clinical presentation, our data suggest that COVID-19 had a severe impact in Lusaka with much loss of life. Second, COVID-19+ deaths were concentrated in the community where testing for COVID-19 was absent. Third, rather than being concentrated in the elderly, COVID-19+ deaths were distributed widely across the age spectrum, and most (~80%) were in individuals aged <60 years. Fourth, COVID-19 was frequently identified among children. Among children ≤5 years, gastrointestinal complaints were common and respiratory symptoms comparatively uncommon, and we suspect that COVID-19 was more likely a coincidental finding in these younger individuals. Finally, the emergence of the Beta and Delta

variants coincided with marked increases in the prevalence of COVID-19+ deaths, reaching ~90% at the peak of those two waves. Conservatively, COVID-19 was identified in 30% of all decedents between January and June 2021.

A consistent finding was the low proportion of decedents with COVID-19 who were diagnosed in life. We hypothesise that this can be explained as follows: (1) the majority of COVID-19+ deaths (~80%) occurred in the community where, with few exceptions, testing was unavailable; (2) among the hospital deaths, only half were tested and (3) most of that testing done at facilities used rapid antigen tests, which are less sensitive than PCR.[27] Putting these in sequence, from the 1118 decedents tested in our study, we identified COVID-19 in 327 (unweighted). Of these, only 25 had been identified antemortem at UTH, or <10% of the total based on PCR. Since these hospital data feed into the national COVID-19 surveillance data, we conclude that the prevalence COVID-19 was systematically and significantly under-represented. While it is certainly possible that individuals in the community might have self-tested for COVID-19 using a rapid test, it is very unlikely that such results would be reported to the government of Zambia and hence fail to be captured in official government reports.

As in our prior analysis, COVID-19+ deaths were concentrated in Lusaka's most densely populated wards where Lusaka's poorest and most vulnerable citizens reside. Access to care is challenging in these communities, and we have described elsewhere how delays in seeking care may contribute to infant deaths in the community.[28] This highlights a sad element to the global COVID-19 narrative. Even within a poor country like Zambia, there is a gradient of impact that appears to fall hardest on those with the least resources.

There are several important distinctions from our findings in round 1. First, the numbers of COVID-19 deaths and the proportion of deceased individuals who tested positive for COVID-19 both increased substantially. This coincided with the emergence of the Beta and Delta variants.

Second, the proportion of these deaths that were judged as 'probably' or 'possibly' due to COVID-19 declined from ~100% in round 1 to ~70% in round 2. We acknowledge that the assessment of causality was limited by the evidence present in the medical chart (facility deaths) or reported through the verbal autopsy (community deaths). With that caveat, our procedures and tools and field team for collecting these data were the same in rounds 1 and 2, leading us to conclude that this was a real shift and not an artefact of shifting study procedures.

Our data do not provide an explanation for this decline, but several explanations are plausible. One is that earlier population exposure to COVID-19 (following the first wave) might have provided increased resistance to the SARS-CoV2 virus, that is, the change is explained by naturally acquired herd immunity from prior exposure. COVID-19 vaccines were virtually unavailable during this period, so vaccine-derived immunity cannot explain this.

A second possibility is that the Beta and Delta variants dominated the second and third waves of transmission. Both variants exhibited increased transmissibility, which plausibly accounts for the high proportions of deaths with COVID-19 at the peak of each wave. As seen with the Omicron variant, a gain in transmissibility could yield a reciprocal loss of virulence, that is, the change is explained by virology.[29 30] However, recent epidemiologic and in vitro data suggest the opposite, that the Beta and particularly Delta variants were more, not less, pathogenic.[31–33] Thus, we the former explanation is more plausible.

## STRENGTHS AND LIMITATIONS

A strength was that our data were collected prospectively, capturing a wide spectrum of ages in community and facility settings and without prior knowledge of clinical presentation or antemortem testing results. While our capacity only allowed us to enrol a subset of all deaths at the morgue, enrolment was conducted without prior knowledge of clinical presentation or PCR status, and thus should not introduce bias affecting overall prevalence estimates after adjusting for weighting. And while all came from the UTH morgue, 80% of all deaths in the city transit this facility, making it highly representative of Lusaka at large. Additionally, we were able to combine multiple data elements on the same individual: clinical presentation, molecular testing, geography and viral lineage, to provide an overall picture of the impact of the COVID-19 pandemic in Lusaka during these periods.

Limitations include that there was a 3-month gap in surveillance between rounds 1 and 2, and that round 2 ended prior to the resolution of wave t3 from the Delta variant. These surveillance periods were defined by funding for the research, not by knowledge of COVID-19 prevalence. Our adjudications were necessarily limited by the completeness of clinical data for the facility deaths and by the accuracy of recall from non-medical persons through the verbal autopsy for community deaths. That could lead to misclassification during our adjudications. Our data describe prevalence but cannot provide a case fatality rate since we lack concurrent incidence data for the general population. Finally, while round 2 data confirm identify a high proportion of paediatric deaths with COVID-19 and again show that gastrointestinal symptoms are common in children<5 years, we are very reluctant to inferring causality in this age group. In our judgement, resolving this question requires a higher gold standard, such as PCR coupled with histopathology. Our data do not address the impact of COVID-19 in terms of a case or infection fatality rate, but this question is being examined in an ongoing excess mortality analysis to be published separately.

The most challenging aspect of this work was inferring the causal role of COVID-19 in each death. We note that the clinical definitions applied during adjudication are themselves of limited precision. However, these are the

same definitions we had used in our original report in *The BMJ 2021*, and we believe our approach here is defensible for several reasons. First, it is supported by guidance from the US CDC: given a patient who presented with respiratory distress and fever and died with a positive PCR test for COVID-19, it is no great leap of faith to infer that the death was probably due to COVID-19. Second, we had conducted MITS on a subset of decedents who had been identified with COVID-19 in life. As previously published, the pathology from these individuals was consistent with known patterns of COVID-19 disease, with evidence of lung and other organ dysfunction.[26] Additionally, we have shown that viral loads were higher among individuals whose deaths were deemed 'probably' due to COVID-19 than those deemed 'probably not' due to COVID-19. And we demonstrated that the ratio of weak to strong PCR signals is age dependent, with much higher ratios in those children and adolescents than older age groups. Combined, these results argue that COVID-19 was a causal factor for deaths in older individuals and more likely a coincidental finding in those ≤19 years. Causality is far less likely for those deemed 'possibly' due to COVID-19. But such cases comprised a small proportion of all COVID-19 positive deaths in this analysis.

## CONCLUSIONS

COVID-19 was highly prevalent in the postmortem setting in Lusaka during the first half of 2021 reaching ~90% of deceased individuals during peak periods of transmission and 30% prevalence over the 6 months of observation. Despite increased rates of antemortem testing for COVID-19, there was little commensurate increase in antemortem testing for those who died in the community, who constituted a 4:1 majority of COVID-19+ deaths. Overall, only about 10% of COVID-19+ decedents had been identified in life, yielding a ~10-fold undercount. Considering recent reports demonstrating examples of under-reporting of COVID-19 elsewhere, including inferential methods from excess mortality analyses,[9–12 34–39] we believe that our results are more likely typical than exceptional. If so, COVID-19's impact in Africa has been underestimated.

## Author affiliations

[1] Department of Global Health, Boston University School of Public Health, Boston, Massachusetts, USA
[2] Biomedical Sciences, University of Zambia, Ridgeway Campus, Lusaka, Lusaka, Zambia
[3] Program for Applied Technology in Health (PATH), Lusaka, Zambia
[4] Avencion Limited, Lusaka, Zambia
[5] Biostatistics and Epidemiology Data Analytics Center, Boston University School of Public Health, Boston, Massachusetts, USA
[6] Macha Research Trust, Choma, Southern Province, Zambia
[7] Pediatric Infectious Diseases, Boston Medical Center, Brookline, Massachusetts, USA
[8] Wildlife Sciences, The Copperbelt University, Kitwe, Copperbelt, Zambia
[9] Avencion, Lusaka, Zambia
[10] Biomedical Sciences, University of Zambia School of Veterinary Medicine, Lusaka, Lusaka, Zambia
[11] Program for Applied Technology in Health, Lusaka, Zambia
[12] Biomedical Sciences, University of Zambia University Teaching Hospital, Lusaka, Lusaka, Zambia
[13] Tufts University School of Medicine, Boston, Massachusetts, USA
[14] University of Zambia School of Veterinary Medicine, Lusaka, Zambia

**Acknowledgements** We wish to thank our team at the Bill & Melinda Gates Foundation for their support and encouragement: Prachi Vora, Padmini Srikanthia and Keith Klugman.

**Collaborators** Not applicable.

**Contributors** CJG was the principle investigator, conceived the project, secured the funding, assisted with the protocol, led on writing the manuscript and acting as guarantor; LM was the Zambian PI, assisted with protocol and tool development and oversaw the field team and study implementation; WBM was the study statistician, developed the analysis plan and conducted the analysis; GK was the technical lead for the PCR laboratory in Lusaka; RCP was the project manager and a research fellow, developed the study proposal, oversaw study management and contributed to key aspects of data analysis and presentation; LE was a research fellow and contributed to key aspects of data analysis and presentation; RN, ChiC and BY were the PCR technicians for our laboratory; SC, LP, GM and DN were the field data collection team under team leaders BN and ChaC (unrelated to ChiC), who conducted grief counselling for the families, obtained consent and collected all of the clinical data and the biological samples: LF supervised creation of the RedCAP data collection tools and oversaw data management, cleaning and retention; RL contributed to data interpretation and analysis; JL provided information technology support to the field team; BK, JM, BM and WM conducted the genetic sequencing analyses under the supervision of DB and ES; ZM was the PCR laboratory manager; NP was a research fellow who supported some of the analyses; MM and NS contributed to the genetic sequencing analyses and DMT was our senior coinvestigator who contributed to data analysis and interpretation.

**Funding** The original ZPRIME study and this COVID-19 expansion were made possible through the generous support of the Bill & Melinda Gates Foundation (OPP 1163027). The funders had no role in designing the study; in the collection and analysis of data or in the decision to submit the article for publication.

**Competing interests** None declared.

**Patient and public involvement** Patients and/or the public were not involved in the design, or conduct, or reporting or dissemination plans of this research.

**Patient consent for publication** Consent obtained from next of kin.

**Ethics approval** The research was conducted with the approval of the ethical review boards from Boston University (reference number H-36469) and the University of Zambia (reference number 015-11-18); written informed consent was obtained from the next of kin or family representative in all cases. **Dissemination of results** Round 2 data were presented to members of the Zambian Ministry of Health and the Zambian Medical Association in July 2021, and the final manuscript was shared with the Zambian National Health Research Authority (NHRA) prior to submission. The results have been presented to our team at the Bill & Melinda Gates Foundation and to representatives from the US CDC and the Swiss Tropical Medicine Institute. These data have been shared with a modelling team at the Imperial College London to support our ongoing excess mortality analyses. Data from round 1 were published previously in the BMJ.

**Provenance and peer review** The lead authors (CJG and LM) affirm that the manuscript is an honest, accurate and transparent account of the study being reported; that no important aspects of the study have been omitted and that any discrepancies from the study as originally planned (and, if relevant, registered) have been explained.

**Data availability statement** Data are available upon reasonable request.

**ORCID iDs**
Christopher J Gill http://orcid.org/0000-0003-3353-0617
William B MacLeod http://orcid.org/0000-0001-8003-8874
Benjamin Mubemba http://orcid.org/0000-0002-5266-3602
Edgar Simulundu http://orcid.org/0000-0001-9423-0816

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
