## [Reviewer comments · BMJ Open]

ARTICLE DETAILS

TITLE (PROVISIONAL)	What is the prevalence of COVID-19 detection by PCR among deceased individuals in Lusaka, Zambia? A post-mortem surveillance study
AUTHORS	Gill, Christopher; Mwananyanda, Lawrence; MacLeod, William; Kwenda, Geoffrey; Pieciak, Rachel; Etter, Lauren; Bridges, Daniel; Chikoti, Chilufya; Chirwa, Sarah; Chimoga, Charles; Forman, Leah; Katowa, Ben; Lapidot, Rotem; Lungu, James; Matoba, Japhet; Mwinga, Gift; Mubemba, Benjamin; Mupila, Zachariah; Muleya, Walter; Mwenda, Mulenga; Ngoma, Benard; Nakazwe, Ruth; Nzara, Diana; Pawlak, Natalie; Pemba, Lillian; Saasa, Ngonda; Simulundu, Edgar; Yankonde, Baron; Thea, Donald

VERSION 1 – REVIEW

REVIEWER	Ghartey-Kwansah, George
REVIEW RETURNED	05-Aug-2022

GENERAL COMMENTS	Reviewer's Comment The overall work is well written and could be a source of literature for the scientific community if the following issues are addressed. 1. The title is misleading as COVID-19 wasn't confirmed as the cause of the various death. This is seen in the classification below: From these syndromic data, we classified adult decedents (>19 years) into the following categories: 1) Probable COVID-19 deaths, where the individual presented with any combination of fever and/or respiratory symptoms; 2) Possible COVID-19 deaths, where the individual presented without classic respiratory symptoms, but rather with known sequelae of COVID-19 suggestive of a heart attack, stroke, or other acute vascular event related to COVID-19's propensity to increase the risk of clotting disorders;¹⁴ 3) Probably not COVID-19 deaths, where the individual had an exonerating cause of death suggesting that the COVID-19 finding on PCR was incidental. 4) Unknown, where the data were insufficient to adjudicate. Yet the authors were emphatic in stating that COVID19 as the cause of most death. The authors claim they have modified via a memo but I haven't seen anything to that effect. 2. Despite increased rates of antemortem testing for COVID-19, there was little commensurate increase in antemortem testing for
---

	those who died in the community, who constituted the a 4:1 majority of all COVID-19 positive deaths. Although, the authors have made some changes to reflect reviewers' comments, they failed to provide evidence of the 4:1 community death caused by COVID19. Having the virus in your system does not mean you died of the virus. 3. Hence, COVID-19 deaths only appeared rare in Lusaka because testing was rarely done. We theorized that the so-called 'Africa Paradox' was a myth born from insufficient surveillance data. Other groups have reached similar conclusions. The decedent having corona virus in the body doesn't mean the death was caused by the virus. They might be asymptomatic at the time of their death and that something else might have caused their death. Evidence to support your theory is very weak to say the least. 4. We defined pediatric deaths as those occurring between 0-≤19 years. I am not expert in public health but is 19 years not a young man or woman? 5. The authors conclude that COVID-19's impact across Africa has been vastly underestimated. How do you use the result of just a city in a country to conclude for the over 55 countries in Africa? The authors are highly biased in their reportage. My question still holds and not convinced with the authors response. In listen examples in other African countries, the authors were still mentioning either capital towns or densely populated cities where high numbers of COVID19 are expected? Why didn't the authors select a few more cities to compare but based on only Lusaka because it is the capital and much more populated. Again, Lusaka is exposed to the outside world than the other cities (eg. Kabwe, Chipata, Ndola etc) and as such high prevalence of COVID19 is expected Although the authors tried to respond to the issues raised, most of them were not attended or partly attended. The work still needs major revision in order to be considered for acceptance.
--	---

REVIEWER	Kirenga, Bruce Makerere University, Department of Medicine and Makerere Lung Institute
REVIEW RETURNED	27-Aug-2022

GENERAL COMMENTS	My comments were addressed however some few more  1. Was autopsy done in addition to taking swabs. If not clearly state so and probably add as limitation 2. Defining children up 18 is a bit unusual, usually 12 years is considered 3. Note this is a single hospital study. Are addresses available to show spread across the country 4. The tables are still many, can they be combined into one table of characteristics etc 5. Also note this is urban data and generalizability may be an issue. 6. Abbreviation CV19 is not common Thanks
---

VERSION 1 – AUTHOR RESPONSE

New comments from the second review:

Reviewer: 1 George Ghartey-Kwansah

Comments to the Author:

Reviewer's Comment

The overall work is well written and could be a source of literature for the scientific community if the following issues are addressed.

1. The title is misleading as COVID-19 wasn't confirmed as the cause of the various death. This is seen in the classification below:

From these syndromic data, we classified adult decedents (>19 years) into the following categories:

1) Probable COVID-19 deaths, where the individual presented with any combination of fever and/or respiratory symptoms;

2) Possible COVID-19 deaths, where the individual presented without classic respiratory symptoms, but rather with known sequelae of COVID-19 suggestive of a heart attack, stroke, or other acute vascular event related to COVID-19's propensity to increase the risk of clotting disorders;¹⁴

3) Probably not COVID-19 deaths, where the individual had an exonerating cause of death suggesting that the COVID-19 finding on PCR was incidental.

4) Unknown, where the data were insufficient to adjudicate.

Yet the authors were emphatic in stating that COVID19 as the cause of most death.

The authors claim they have modified via a memo but I haven't seen anything to that effect.

*** In this second revision, we have revised the title of the paper to make clear that we are reporting the prevalence of CV19 detection rather than asserting causality.

The new title is: What is the prevalence of COVID-19 detection by PCR among deceased individuals in Lusaka, Zambia? a systematic post-mortem study

2. Despite increased rates of antemortem testing for COVID-19, there was little commensurate increase in antemortem testing for those who died in the community, who constituted a 4:1 majority of all COVID-19 positive deaths.

Although, the authors have made some changes to reflect reviewers' comments, they failed to provide evidence of the 4:1 community death caused by COVID19. Having the virus in your system does not mean you died of the virus.

*** We have revised the paper extensively to express more caution about asserting causality. As to this section in the paper specifically, this has been revised to be more circumspect as follows, "After weighting to adjust for the differing enrollment ratios, community deaths accounted for 79% and facility deaths for 21% of total COVID-19+ deaths. Stated another way, for each COVID-19+ decedent at a facility, ~four COVID-19+ decedents were identified in the community."

Here we removed the term 'death' which seemed to imply causality to the reviewer and replaced it with 'decedent' referring simply to the ratio of detections in the community vs. facility settings, which is unambiguous. Please see page 10 of the revision.

3. Hence, COVID-19 deaths only appeared rare in Lusaka because testing was rarely done. We theorized that the so-called 'Africa Paradox' was a myth born from insufficient surveillance data. Other groups have reached similar conclusions.

The decedent having corona virus in the body doesn't mean the death was caused by the virus. They might be asymptomatic at the time of their death and that something else might have caused their death. Evidence to support your theory is very weak to say the least.

*** We have addressed this limitation throughout the revised paper.

4. We defined pediatric deaths as those occurring between 0-≤19 years.

I am not expert in public health but is 19 years not a young man or woman?

*** This question was repeated verbatim from the original review in July:

“I am not expert in public health but is 19 years not a young man or woman?”

Our previous response in the July review was:

“This was an arbitrary cut-point. There was only one person who was 19 in this age group, so it would not make any difference if we limited to 18 years. However, we found it efficient to divide the cohort up into 20-year age strata and doing this different from one age group did not make much sense to us. So, we chose 0-19 years. We do not think that was unreasonable.”

I believe that we have responded already to this comment. The distinction between 18 vs. 19 years is arbitrary and since there was only 1 person who was 19, the impact is nil. What is important is that we had good reason to divide the cohort into 20-year increments. We stand by that decision.

5. The authors conclude that COVID-19's impact across Africa has been vastly underestimated.

How do you use the result of just a city in a country to conclude for the over 55 countries in Africa?

The authors are highly biased in their reportage.

***With respect, this is how the concluding paragraph had been re-worded in the July revision:

“Considering recent reports demonstrating examples of underreporting of COVID-19 elsewhere, including inferential methods from excess mortality analyses, we believe that our results are more likely typical than exceptional. If so, COVID-19's impact in Africa has been underestimated.” (I have bolded elements that were included to convey how we reached this conclusion and that are therefore offering a qualified opinion).

First, we clarify that our opinion is based on data from our study and from what we have learned elsewhere through excess mortality analyses and seroprevalence studies across Africa. For example, please see the excellent analysis posted to MedRxiv by Mark Lipsitch and colleagues at Harvard School of Public Health that reports a significant undercount of CV19 deaths across Africa. This meta analysis included our post mortem prevalence data from the BMJ 2021 paper, plus dozens of other African countries based on sero- and molecular surveys. In some cases, the undercount from official reports vs. serosurveys was astonishing (>1000 fold in Malawi). (see <https://doi.org/10.1101/2022.07.03.22277196>).

We wish to emphasize from the Lipsitch analysis that they were able to locate ONLY one source of data about CV19 prevalence in the post mortem setting: that was our work in Lusaka.

Similarly, Lewis et al recently published a comprehensive systematic analysis and meta-analysis of seroprevalence data from Africa that was published in BMJ-Global Health in the summer of 2022. It reached the same conclusion.

Such reports are part of the basis for our interpretation of our data and the basis for our stated opinion in the discussion. In other words, our conclusion that our results are likely the tip of the iceberg are being confirmed by other groups.

Second, the revised language we offered in July does not seem to be egregiously overstating our conclusions. We tried to be quite circumspect. “Considering recent reports. . . we believe our results are more likely typical than exceptional.” and, “. . . if so, COVID-19's impact in Africa has been underestimated.” By including such caveats we sought to emphasize that this was our conclusion and also our opinion. Stating opinions is appropriate in a discussion section. The reviewer is entitled to a different opinion, but we are entitled to declare our opinions in a discussion section.

My question still holds and not convinced with the authors response. In listen examples in other African countries, the authors were still mentioning either capital towns or densely populated cities where high numbers of COVID19 are expected? Why didn't the authors select a few more cities to compare but based on only Lusaka because it is the capital and much more populated.

*** We only had a team and funding to do this study in Lusaka. We did not have a team in Ndola, Livingstone, Kitwe or elsewhere. The reviewer is unhappy that we could not do more than we did. We also wish we had been able to do more. But we had a trained research team in Lusaka and

a molecular lab and trained technicians and funding to do a study in Lusaka. We did not have funds to do a multi city, multi country study.

It is important to emphasize that setting up a field site like this takes a lot of time and resources. For this work, it took us nearly a year to establish the infrastructure in Lusaka that allowed us to do the prior ZPRIME study (which we recently published in Lancet Global Health). We had to hire and train staff and set up a field lab and hire and train them too, and then create a protocol and get it approved by two IRBs under the supervision of an external scientific advisory committee (that included Dr. Madhi). It was because of ZPRIME and that pre-existing infrastructure that we could pivot so quickly and study the emergent COVID outbreak in Zambia when the pandemic hit so abruptly. If we had not already established that infrastructure, we would not have had time to set up such a study de novo in Lusaka either. Based on what we experienced building ZPRIME, it would have taken a year or more to set up infrastructure in other countries (where we have never worked before and have no established capacity or network of collaborators). By the time we could have created that capacity, the opportunity to study CV19 would have been lost. Epidemics do not wait while you create the perfect infrastructure and we were fortunate to be in the right place at the right time and to be able to react as quickly as we did. We can only report on the study that we did, not the study we wish we had had the time and resources to do. Thus, I do not find this comment very helpful because it is not actionable after the fact.

Again, Lusaka is exposed to the outside world than the other cities (eg. Kabwe, Chipata, Ndola etc) and as such high prevalence of COVID19 is expected

*** While this is obviously true, the same argument could be made for Nairobi, Kinshasa, Cairo, Johannesburg, Accra, Dhaka, Dar es Salaam – or London. Is the reviewer suggesting that impact studies should not be done in urban centers, because they are globally interconnected? Every big city is part of a global network. And in Zambia, the capital Lusaka is the largest city and where nearly 20% of the total population of Zambia resides. Being the largest population center in Zambia, Lusaka is representative by definition.

The impact of infectious diseases is always influenced by the population structure and social and geographical networks within that place. According to the most recent report from the Organization for Economic Co-operation and Development (OECD), 50% of all Africans now live in cities, and the urbanization trend is accelerating (see https://read.oecd-ilibrary.org/development/africa-s-urbanisation-dynamics-2020_b6bccb81-en#page143/a). Thus, the impact of a disease like COVID19 in urban settings is more typical of CV19's behavior in Africa than exceptional.

Reviewer: 2

Dr. Bruce Kirenga, Makerere University

Comments to the Author:

My comments were addressed however some few more

1. Was autopsy done in addition to taking swabs. If not clearly state so and probably add as limitation

***This question also made us concerned that the reviewer had not seen our revised MS or the response memo. The reason for that concern is because in our earlier response memo, we had stated that minimally invasive tissue sampling WAS done and that these results had previously been published. That was noted in the July response memo on pages: 2, 8, 11 (twice), 12 and 15, and the reference to the Mudenda et al paper in CID was provided twice. In the revised MS itself, this issue was discussed on pages 13 and 18, and we again directed the reader to the Mudenda et al CID paper in text (see ref 24).

2. Defining children up 18 is a bit unusual, usually 12 years is considered

*** Again, this was already addressed in the July memo and revision. We point to our earlier response.

'This was an arbitrary cut-point. There was only one person who was 19 in this age group, so it would not make any difference if we limited to 18 years. However, we found it efficient to divide the cohort up into 20-year age strata, and doing this different from one age group did not make much sense to us. So, we chose 0-19 years. We do not think that was unreasonable.'

3. Note this is a single hospital study. Are addresses available to show spread across the country

*** We do not have addresses, and this would be considered identifiable information and disallowed by an IRB. What we provide instead is the city wards (i.e., neighborhoods) where the decedents were residing in Lusaka, and these were presented previously in Figure 2 (no change from original MS). These were all deaths that had occurred in Lusaka. We have no data about the rest of Zambia.

4. The tables are still many, can they be combined into one table of characteristics etc

*** There are only four in-text tables in the paper. Each tells a different story that follows the narrative of the results, and each table answers a different question. In my opinion, it makes sense to leave them as they are.

Table 1 presents PCR results by setting (facility vs. community)

Table 2 presents PCR results by demographic features

Table 3 presents PCR results by age strata

Table 4 presents the first part of the causal inference analysis (the second half is in Supp Table 3)

5. Also note this is urban data and generalizability may be an issue.

*** We have addressed the issue of generalizability at several points in the paper.

6. Abbreviation CV19 is not common

*** We have harmonized to use 'COVID-19' throughout, with the exception of where we name the virus specifically, which is SARS-CoV-2.